# Morphological and Molecular Identification of Hard Ticks in Hainan Island, China

**DOI:** 10.3390/genes14081592

**Published:** 2023-08-06

**Authors:** Jitrawadee Intirach, Xin Lv, Qian Han, Zhi-Yue Lv, Tao Chen

**Affiliations:** 1Hainan General Hospital, Hainan Affiliated Hospital of Hainan Medical University, Haikou 570100, China; intirach@163.com; 2Laboratory of Tropical Veterinary Medicine and Vector Biology, School of Life Sciences, Hainan University, Haikou 570228, China; qianhan@hainanu.edu.cn; 3International School of Public Health and One Health, Hainan Medical University, Haikou 571199, China; lvxin@hainmc.edu.cn; 4Provincial Engineering Technology Research Center for Biological Vector Control, Guangzhou 510080, China; 5Key Laboratory of Tropical Disease Control, Ministry of Education, Sun Yat-Sen University, Guangzhou 510080, China; 6Hainan Provincial Bureau of Disease Prevention and Control, Haikou 570100, China

**Keywords:** *cox1* gene, *16S* rRNA gene, genetic diversity, hard tick

## Abstract

Ticks are small, blood-sucking arachnids, known vectors of various diseases, and found throughout the world. They are distributed basically in almost all regions of China. At present, there is not much information regarding tick species on Hainan Island. They were subjected to morphological identification and imaging on an individual basis. Molecular phylogenetic analyses, based on *cox1* and *16S* rRNA genes, were utilized to identify the species and determine their approximate phylogenetic origin and genetic diversity. The genomic DNA of tick species was extracted, and cytochrome oxidase subunit 1 (*cox1*) and *16S* ribosomal RNA (rRNA) genes were amplified and sequenced. The identification of five tick species, namely *Rhipicephalus microplus*, *Rhipicephalus sanguineus*, *Rhipicephalus haemaphysaloides*, *Haemaphysalis cornigera* and *Haemaphysalis mageshimaensis*, was carried out by morphological analysis. When employing the *cox1* and *16S* rRNA phylogenetic tree, all isolates of *R. microplus* from Hainan Island were classified as clade A and B, respectively. *R. sanguineus* was recognized as a member of the tropical lineage by phylogenetic analysis on the *cox1* and *16S* rRNA genes. Three phylogenetic groups of *R. haemaphysaloides* were recognized and found to be related closely to strains from China. *H. cornigera* and *H. mageshimaensis* formed one phylogenetic group, presumably from tick strains prevalent in Japan and China. The haplotype network analysis indicated that *R. microplus* is classed into 26 and 6 haplotypes, which correspond to *cox1* and *16S* rRNA gene assemblages, respectively. In addition, four *cox1* haplotypes were detected in *R. sanguineus*. This is the first evidence that suggests genetic diversity, host range and geographical distribution of hard ticks in Hainan Island, China.

## 1. Introduction

Ticks are blood-sucking ectoparasitic arthropods that play an essential role in transmitting various pathogens to humans and animals worldwide [1,2], they are the second most significant transmitters of infectious diseases after mosquitoes [3,4]. Most of the pathogenic organisms that inhabit ticks include viruses, bacteria (especially rickettsiae and spirochetes), protozoa and helminths, with their diversity increasing over the past three decades [5,6]. Tick-borne illnesses have been reported in nearly all Provinces/Autonomous Regions/Municipalities (P/A/M) in China in recent years, and the incidence rate is on the rise [7,8]. Roughly 800 tick species and 18 genera have been recognized globally, whereas, 9 genera have been identified in China, with 111 and 14 species of hard (Ixodidae) and soft ticks (Argasidae), respectively [9,10]. Morphological identification refers to the process of visually identifying tick species based on their morphological characteristics, such as body shape, size, color and pattern [11,12]. This involves the use of specialized keys or taxonomic guides that provide detailed descriptions and illustrations of different tick species, as well as their geographic distribution, host preferences and pathogen transmission potential [13]. However, morphological identification is insufficient in separating between related species complexes, particularly when the specimens are at immature stages, physically damaged or engorged [14]. Therefore, molecular techniques such as sequencing of the cytochrome oxidase subunit 1 (*cox1*) gene and *16S* ribosomal RNA (*16S* rRNA) gene can be used to characterize tick species [15,16]. The *cox1* gene is used commonly for species identification and phylogenetic studies [17], while the *16S* rRNA gene is a useful marker for determining the evolutionary relationships between different tick species [18]. These techniques can be used in combination with morphological identification to provide a more accurate and comprehensive tick species characterization [19,20].

Hainan Island is located in the southernmost part of China and it has a hot climate. Climatic conditions have shaped rich animal and plant resources on the island and provided a suitable environment for the reproduction of ticks. Previous research on tick species distribution on domestic and small wild animals in Hainan has been investigated [21,22]. However, these studies were conducted over a long period of time, in which climate change and changes in the range of ticks occurred, and no update on data for ticks has been carried out in Hainan Island in the past ten years. At present, there is little information regarding tick species on Hainan Island. Therefore, this study conducted a survey on the distribution of ticks in Hainan Island from July to December 2022 in order to report the morphological features and genetic diversity of field-collected ticks within hard ticks collected from dogs, goats and cattle in the area. Differentiations of hard ticks were characterized using morphological features and nucleotide analysis of the *cox1* mitochondrial gene and *16S* rRNA gene.

## 2. Materials and Methods

### 2.1. Study Sites and Tick Collection

Tick samples were collected in Hainan Island, China, from July to December 2022. Hainan Island is located in southern China, and it has a hot climate, sufficient sunshine and no snow throughout the year. Its climate is mostly tropical with between 1500 and 2000 millimeters of average annual precipitation and average temperature ranging from 23 to 26 °C, and it sits on a geographical coordinate of 20°1′ N latitude and 110°20′ E longitude. On the island, a total of 858 adult and nymph ticks were collected from cattle, dogs and goats in 24 sites and 12 districts of cowshed farms, dog houses and farms and goat farms, respectively (Figure 1 and Appendix A). Sampling points, geographical coordinates, tick samples, hosts, habitat and time were recorded. Ticks were collected directly from animal hosts by using steel forceps, before placing them in sterile tubes for transportation to the laboratory at low temperature. Then, they were rinsed 3 times with 70% ethanol, rinsed once with phosphate-buffered saline (PBS), air dried and kept at −80 °C until further processing.

### 2.2. Morphological Identification and Photography

Based on morphological characteristics, the ticks were recognized under a light stereomicroscope Leica S9E (Leica Microsystems, Heerbrugg, Switzerland), and their species were identified using a previously reported taxonomic key [13,23]. The genus, species, stage and gender of the ticks were sorted and counted. Out of the 858 ticks examined, 199 (2–30 specimens per site) were sampled and photographed, with a total of 15,000 pictures taken with a stereomicroscope Leica MZ10F connected to a computer, and the processing and measurement of the images were carried out using Leica Application Suite X version 5.0 software (Leica Microsystems, Heerbrugg, Switzerland). All of the selected ticks were observed, and the following characteristics were taken: dorsal whole-body view; ventral whole-body view; coxae; scutum; dorsal capitulum view; ventral capitulum view; genital opening; anal aperture; spiracular plates. Approximately 5–15 captures were taken in each position.

### 2.3. DNA Extraction and PCR Amplification of the Tick cox1 and 16S rRNA Genes

After morphological study, 199 selected specimens from different areas, hosts and tick species were subjected to molecular study. The QIAamp Blood and Tissue Kit (Qiagen, Hilden, Germany) was utilized in accordance with the manufacturer’s protocol to perform DNA extraction. Individual ticks were rinsed with PBS and air dried for 5 min on sterile paper. Only semi and fully engorged ticks were sliced separately into small pieces along the length of the body by a sterile scalpel blade. They were then placed in a 2 mL microcentrifuge tube filled with glass beads (900 mg, 0.1 and 3 mm in diameter), together with 180 microliters of ATL buffer and 20 µL of proteinase K, before homogenizing while shaking for 3 minutes in a tissue grinder at 60 HZ for extracting DNA. The extracted DNA was frozen at −20 °C until used.

Individual specimens for each tick species were subjected to polymerase chain reaction (PCR) to amplify regions of the *cox1* and *16S* rRNA genes. PCR conditions and primers have been summarized in Appendix A. The PCR reaction mix (final volume 25 µL) included 12.5 µL of 2X TaKaRa Ex Premier™ DNA Polymerase Dye plus (Takara, Japan), 1.5 µL of 5 µM forward and reverse primers, 1–4 µL of DNA template (approximately 100 ng of genomic DNA) and 5.5–8.5 µL of ddH_2_O. A negative no-template water control was included in all of the PCR runs. Each amplified product was loaded onto 1.5% agarose gel (Biowest, Shanghai, China) using five microliter volumes per sample and the resulting bands were visualized on a gel documentation system (JUNYI, Beijing, China). After purification using the QIAquick^®^ PCR Purification Kit (Qiagen, Hilden, Germany), by following the manufacturer’s instructions, the PCR amplified products were sent subsequently to the Sangon Company (Guangzhou, China) for Sanger bidirectional sequencing.

### 2.4. Phylogenetic Analyses

Sequences of *cox1* and *16S* rRNA genes were edited manually and assembled, and bidirectional consensus sequences were originated using BioEdit software version 7.2.6.1 [24]. The sequences were compared with those available in GenBank using the Basic Local Alignment Search Tool (BLAST) available at the National Center for Biotechnology Information (NCBI) website “https://blast.ncbi.nlm.nih.gov/Blast.cgi (accessed on 4 March 2023)”. The multiple sequences were performed using the default parameters of the ClustalW multiple alignment tool in MEGA 11. The phylogenetic analysis was performed using the maximum likelihood (ML) or neighbor-joining (NJ) methods, based on the general time reversible (GTR) model [25] or Kimura’s two-parameter model (K2P) [26] in MEGA 11 [27]. Bootstrap values were estimated for 1000 replicates. The phylogenetic trees were edited using adobe illustration (AI) software.

### 2.5. DNA Polymorphism Analysis

DNA sequences from each tick species and gene fragment (*cox1* and *16S* rRNA) were defined as sequence sets that estimated the genetic differentiation and gene flow by using DnaSP software version 6.12.03 [28]. The number of sequences (*N*), number of haplotypes (*Hn*), number of segregating sites (*n*), haplotype diversity (*Hd*), average number of nucleotide differences (*k*), nucleotide diversity (*π*) and *F_ST_* values between pairs of populations (pairwise *F_ST_*) were generated. The haplotype networks were created to evaluate the ancestral relationship between detected haplotypes using the Median Joining Network [29] in PopART software version 1.7 [30].

## 3. Results

### 3.1. Morphological Features of Tick Species

A total of 858 ticks were collected from 94 domestic animals, including 74 cattle, 11 dogs and 9 goats (Appendix A). Most of the ticks collected from hosts were in the adult stage (n = 720), from which 550 were female (76.39%) and 170 male (23.61%). There were 138 female nymphs (16.08%) from the total number of samples. The percentage of unfed, semi-engorged and fully engorged ticks was found to be 25.41% (n = 218), 22.84% (n = 196) and 51.75% (n = 444), respectively. The tick species with the highest prevalence was *Rhipicephalus microplus* 70.62% (n = 606), followed by *R. sanguineus* sensu lato (s.l.) tropical lineage (*R. linnaei*) 28.79% (n = 247), *R. haemaphysaloides* 0.35% (n = 3), *H. cornigera* 0.12% (n = 1) and *H. mageshimaensis* 0.12% (n = 1). *R. sanguineus* (*R. linnaei*) ticks were collected from dogs, *R. haemaphysaloides*, *H. cornigera* and *H. mageshimaensis* from cattle and *R. microplus* from cattle and goats. Morphologically, five species of ticks belonged to two genera, namely *Rhipicephalus* and *Haemaphysalis*. The capitulum (mouthparts) is the anterior portion of the body used to distinguish between these genera. It is angulated on the sides with triangular porose areas in the genus *Rhipicephalus*. The palpi are short and broad with no transverse ridges (Figure 2A–F). The capitulum of the genus *Haemaphysalis* is not angulated on the sides, but has large porose areas that are longitudinal and distant. The palpi is short and its second segment has an acute outward basal prolongation (Figure 2G–J). The distinguishing morphological features of *R. microplus*, *R. sanguineus* (*R. linnaei*), *R. haemaphysaloides*, *H. cornigera* and *H. mageshimaensis* are presented in Appendix A. The tick species in this study could be distinguished based on color, body, capitulum, coxae, genital opening, scutum (female), anal plates (male), accessory adanal plates (male) and spiracular plate shapes (Appendix A).

### 3.2. Molecular Identification and Classification of Ticks by Nucleotide BLAST

A total of 199 specimens: *R. microplus* (n = 152), *R. sanguineus* (*R. linnaei*) (n = 42), *R. haemaphysaloides* (n = 3), *H. cornigera* (n = 1) and *H. mageshimaensis* (n = 1) were used. All nucleotide sequences obtained in this study were submitted to GenBank with Accession Numbers: OQ704485‒OQ704683 and OQ725381‒OQ725579 for *cox1* and *16S* rRNA, respectively. Partial *cox1* and the *16S* rRNA DNA sequence of all ticks were 96–100% of sequence matching identities in GenBank using BLAST (Appendix A). The closest sequence to those of *R. microplus*, based on *cox1* genes of 99–100% identity, was from Columbia (KT906178.1), Haikou, China (MK685985.1), Kenya (KX228549.1) and Benin (MT249801.1), and to those of *R. sanguineus* (*R. linnaei*) was from Angola (MF425995.1). The closest sequence to those of *R. haemaphysaloides* with 96% identity, *H. cornigera* with 99% identity and *H. mageshimaensis* with 99% identity, based on *cox1* genes, was from Yingtan, China (OP050242.1), Ganzhou, China (OM368283.1) and Haikou, China (NC062163.1), respectively. The closest sequence to those of *R. microplus*, based on *16S* rRNA genes of 99–100% identity, was from Thailand (KC170742.1) and Mozambique (EU918187.1). The closest sequence to those of *R. sanguineus* (*R. linnaei*) with 100% identity, *R. haemaphysaloides* with 98% identity, *H. cornigera* with 99% identity and *H. mageshimaensis* with 98% identity, based on *16S* rRNA genes, was from Thailand (KC170744.1), Taiwan (AY972533.1), Ganzhou, China (OM368283.1) and Haikou, China (NC062163.1), respectively.

#### 3.2.1. Genetic Distances and Phylogenetic Analyses for *R. microplus*

Genetic analysis of *R. microplus* complex comprises 5 taxa, including *R. annulatus*, *R. australis* and *R. microplus* Clade A, B and C, based on the *cox1* gene [15]. Clade A comprised *R. microplus* from Colombia, Kenya, South Africa, Brazil, Thailand, etc., Clade B contained *R. microplus* from China, and Clade C consisted of *R. microplus* from India, Pakistan, Bangladesh and Myanmar. Clade A and C exhibited a sibling association with *R. australis* and *R. annulatus*, respectively, indicating a close relationship. In this study, all of the *R. microplus* ticks and five isolates from Colombia, China, Kenya, Brazil and Thailand were clustered within Clade A (Figure 3). There was variation in the mean K2P distances between and within groups from 0.050 to 0.104 and 0.000 to 0.016, respectively (Appendix A).

The *16S* rRNA genes revealed 4 taxa of *R. microplus* complex comprising *R. australis*, *R. annulatus* and *R. microplus* Clade A and B. The *R. microplus* isolates from India, Pakistan and China formed Clade A, while those from this study were Clade B, which clustered together with Thailand, China, Taiwan, Malaysia, Africa and South America (Figure 4). The mean K2P distances between and within groups varied from 0.012 to 0.048 and 0.000 to 0.012, respectively (Appendix A).

#### 3.2.2. Genetic Distances and Phylogenetic Analyses for *R. sanguineus*

The *cox1* and *16S* rRNA phylogenetic trees of *R. sanguineus* revealed three genetic lineages, including tropical, temperate and southeast European lineages. Both genes of all the *R. sanguineus* ticks in this study were clustered into one clade of tropical lineage, and closely related to those from Thailand, China, India, Cuba, Brazil, Iraq, Angola, Pakistan and Sub-Saharan Africa (Figure 5). *R. sanguineus* ticks in the southeast European lineage showed a sister relationship with the tropical lineage and were closer genetically to *R. turanicus*, while those in the temperate lineage formed a distinct clade away from the tropical lineage. There was variation in the mean K2P distances between and within groups of *cox1* and *16S* rRNA, ranging from 0.090 to 0.121 and 0.030 to 0.071 and 0.000 to 0.055 and 0.000 to 0.027, respectively (Appendix A).

#### 3.2.3. Genetic Distances and Phylogenetic Analyses for *R. haemaphysaloides*

The phylogenetic tree of *R. haemaphysaloides*, based on *cox1* and *16S* rRNA genes, consisted of three groups. Group 1 comprised *R. haemaphysaloides* ticks from China (Hunan, Sichuan and Yunnan) and Thailand, which were related closely to *R. sanguineus* and *R. turanicus*. Group 2 consisted of ticks from Pakistan, India and Sri Lanka, and Group 3 included ticks from this study and China (Taiwan, Yangxin, Yingtan, Ganzhou and China–Myanmar border) (Figure 6). There was variation observed in the mean K2P distances between and within groups of *cox1* and *16S* rRNA, with a range of 0.086 to 0.150 and 0.055 to 0.073, and 0.000 to 0.038 and 0.000 to 0.049, respectively (Appendix A).

#### 3.2.4. Genetic Distances and Phylogenetic Analyses for *Haemaphysalis* spp.

*H. cornigera* and *H. mageshimaensis* ticks, and four sequences of *cox1* and *16S* rRNA genes from this study formed a single clade and were analyzed for phylogenetic trees (Figure 7). According to phylogenetic analysis, based on *cox1* genes, *H. cornigera* and *H. mageshimaensis* ticks in this study were clustered together in Ganzhou, China and Haikou, China, respectively (Figure 7A). The mean K2P distances between and within species varied from 0.000 to 0.003 and 0.000 to 0.003, respectively (Appendix A). Similarly, *16S* rRNA genes of the *H. cornigera* ticks in this study were clustered alongside the isolates from Japan and Ganzhou, China, while *H. mageshimaensis* ticks in this study were clustered together with those collected from Japan and China (Taiwan and Haikou) (Figure 7B). The mean K2P distances for the *cox1* gene was 0.005, and they varied from 0.000 to 0.003 for the *16S* rRNA gene (Appendix A).

### 3.3. Genetic Diversity and Haplotype Analyses

A total of 398 *cox1* and *16S* rRNA gene sequences from five tick species were analyzed (Appendix A). Twenty-six haplotypes of the *cox1* sequences and 6 of the *16S* rRNA sequences were identified from the 17 populations of *R. microplus* ticks in Hainan Island, with a haplotype diversity (Hd) of 0.840 and 0.109, respectively. Fifty-two sequences of each gene from the six populations of *R. sanguineus* (*R. linnaei*) ticks were analyzed, and four haplotypes of the *cox1* sequences were detected with a low haplotype diversity (*Hd* = 0.216). In contrast, the *16S* rRNA sequences showed no variation site and they generated only 1 haplotype, while two, one and one haplotype were identified from 1 population of *R. haemaphysaloides*, *H. cornigera* and *H. mageshimaensis* ticks, respectively (Appendix A). From this study, only *R. microplus* ticks, based on *cox1* genes, generated a higher haplotype diversity with the *Hd* ranging from 0.200 to 1.000 (mean = 0.840) and nucleotide diversity (*π*) ranging from 0.00171 to 0.00389 (mean = 0.00334). The pairwise *F_ST_* values among populations ranged from –0.5128 to 0.822 (Appendix A). In 136 comparisons, 86 showed a significantly higher genetic differentiation (*p* = 0.05).

The median-joining haplotype network, based on *cox1* and/or *16S* rRNA sequences of *R. microplus* and *R. sanguineus* (*R. linnaei*) in each population is shown in Appendix A. In the frequency haplotype analysis of *R. microplus* ticks, based on *cox1* genes, H1 was the most frequent haplotype, represented by 40 from 142 sequences (28.2%) and a shared haplotype with 13 geographic populations. H9 was positioned centrally on the network map, while the rest of the shared haplotypes were clustered around their respective centers (Appendix A). H1 was the most frequent haplotype detected in the *16S* rRNA sequences, represented by 134 from all sequences (94.4%), while the remaining shared haplotypes formed small cluster centers (Appendix A). H1 was also the most frequent haplotype of *cox1* sequences in *R. sanguineus* (*R. linnaei*) ticks, represented by 46 from 52 sequences (88.5%), while the remaining shared haplotypes formed small cluster centers (Appendix A). The high percentage of haplotype frequency in the *16S* rRNA sequences of *R. microplus*, and *cox1* sequences of *R. sanguineus* (*R. linnaei*), suggested that these populations were relatively stable and capable of adapting to a variety of environments. CLHK was the locality of *R. microplus* ticks, with the highest number of haplotypes (*Hn* = 8 for *cox1* and *Hn* = 3 for *16S* rRNA sequences), while the locality of BWJ, XTDZ and QZ had an equal number of haplotypes (*Hn* = 2) for *R. sanguineus* (*R. linnaei*) *cox1* sequences.

## 4. Discussion

The *R. microplus* species had the highest prevalence in this study and is found in many parts of the world, including South and Central America, Africa, Asia and Australia [31,32]. It is a major livestock pest, particularly affecting cattle. The *R. microplus* complex, based on *cox1* and *16S* rRNA genes, includes 5 and 4 distinct taxa, respectively, including *R. annulatus*, *R. australis* and *R. microplus* (clade A–C for the *cox1* gene and clade A–B for the *16S* rRNA gene) [15,31,33]. However, the *cox1* gene has a greater intraspecies resolution within the *R. microplus* complex when compared to that of the internal transcribed spacer 2 (ITS2), *16S* and *12S* genes [15,33]. A previous study revealed that, apart from *R. australis*, all the other taxa were detected in China [20]. However, in this study the *R. microplus* ticks in Hainan Island were clustered within clade A and B, based on *cox1* and *16S* rRNA genes, respectively, together with isolates from Asia, Africa and South America [32]. *R. sanguineus* complex, based on molecular markers, is classified into four potential lineages, including temperate (*R. sanguineus* s.s.), southeastern Europe, tropical (*R. sanguineus* s.l.) and Afrotropical (*R. afranicus*) [34,35,36,37]. Šlapeta et al. (2021) [37] confirmed that *Rhipicephalus linnaei*, which is synonymous with *R. sanguineus*, was regarded as the oldest name and suggested that its adoption should refer to *R. sanguineus* sensu lato as the “tropical lineage”. In addition, Šlapeta et al. (2022) [38] elucidated further regarding the recognition of the tropical lineage (*R. sanguineus* s.l.) and officially designated it as *Rhipicephalus linnaei*. In this study, the phylogenetic analysis of *cox1* and *16S* rRNA genes identified *R. sanguineus* in Hainan Island as being clustered into one clade of tropical lineage in the temperate zone of a country (Thailand, China, India, Cuba, Brazil, Iraq, Angola, Pakistan and Sub-Saharan Africa), similar to previous reports [15,39,40]. A study by Zemtsova et al., 2016 [41] revealed that *R. sanguineus* of tropical lineage occurs in geographical areas with an annual mean temperature of greater than 20 °C, whereas the temperate lineage presents in geographical areas with an annual mean temperature of less than 20 °C. This supports the presence of *R. sanguineus* in this study site, where the average temperature ranges from 23 to 26 °C. The genetic diversity of *R. haemaphysaloides*, based on the *cox1* gene, had not been investigated previously. Three genetic groups were recognized, based on this gene, and *R. haemaphysaloides* from this study was found to belong to one of the three. The result for the *16S* rRNA gene was similar to that in a previous study by Li et al. (2018) [20], who recognized three genetic groups, from which *R. haemaphysaloides* was found in group one that belonged to the isolated tick from China. This study found two tick samples in the genus *Haemaphysalis* in Changliu town, Haikou district, namely *H. cornigera* and *H. mageshimaensis*, which were collected from cattle. A review of the geographical distribution of ticks in China found that *H. cornigera* was distributed in Fujian, Taiwan, Guangdong, Guangxi and Hainan, and the natural host of this species was buffaloes and cattle [10]. Moreover, a report by Doi et al. (2020) [42] revealed that *H. cornigera* is the dominant tick in wildlife (sika deer) on Niijima Island, Japan. The natural hosts for *H. mageshimaensis* are livestock, wild mammals and birds, which have been reported in China [10,43] and Japan [44]. Although there is little information on nucleotide sequence in BLAST searches and limited determination of evolutionary relationships of these species, the phylogenetic analysis in this study revealed that both *H. cornigera* and *H. mageshimaensis* were clustered and related together with isolates from China and Japan, respectively, with the Chinese isolates based on *cox1* and *16S* rRNA genes.

Moreover, this study also investigated the genetic diversity of different tick species in local areas. The *cox1* gene marker presents greater resolution than the *16S* rRNA gene marker in revealing the genetic variability of *R. microplus* and *R. sanguineus* (*R. linnaei*) in Hainan Island. From the *16S* rRNA gene sequences, only six haplotypes were identified from *R. microplus* ticks in this study, while no variation site in *R. sanguineus* (*R. linnaei*) was observed, and only one haplotype generated. On the other hand, *cox1* gene sequences revealed 26 and 4 haplotypes of *R. microplus* and *R. sanguineus* (*R. linnaei*), respectively. However, only *R. microplus* ticks generated a higher haplotype diversity, based on the *cox1* gene. Similar to previous studies, several COI (*cox1*) haplotypes were revealed with high genetic differences in *R. microplus* [15], and the haplotype frequency in *R. microplus* was higher than in *R. sanguineus* [15,16]. Low et al. (2018) [45] revealed four COI haplotypes with slightly higher variation in *R. sanguineus* in Malaysia, but *16S* rRNA sequences were highly conserved, which was similar to *R. sanguineus* (*R. linnaei*) in Hainan Island in this study. In the median-joining haplotype network, H1 was the most frequent haplotype for *R. microplus* and *R. sanguineus* (*R. linnaei*), based on *16S* rRNA and *cox1* sequences, respectively, with only a few shared haplotypes of all geographic populations in Hainan Island. This suggested that haplotype H1 is an ancestor of *R. sanguineus* (*R. linnaei*) and *R. microplus* and other haplotypes were shared over time. In contrast, the *cox1* haplotype of *R. microplus* from seventeen sampled populations had high genetic diversity and revealed highly abundant shared haplotypes. It is indicated that ticks may have the ability to adapt to different environments and there was frequent gene communication between individuals or populations [46]. However, the remaining three tick species from this study were not investigated, due to the smaller number of tick specimens and species being represented by only one population.

## 5. Conclusions

This study employed morphological and molecular approaches to characterize and identify the genetic relationships of field-collected ticks from eight hundred and fifty-eight hard ticks collected from cattle, dogs and goats in 24 locations around Hainan Island. Based on morphological features, five tick species were identified, namely *Rhipicephalus microplus*, *R. sanguineus* (*R. linnaei*), *R. haemaphysaloides*, *Haemaphysalis cornigera* and *H. mageshimaensis* and confirmed subsequently by molecular analysis of *cox1* and *16S* rRNA genes. It was concluded that *cox1* and *16S* rRNA genes were useful markers for verifying species identification of hard ticks, according to analyses of the sequence data obtained. The analyses established a dependable DNA reference database that could be utilized in forensic entomology not only in China but also in other countries where these species are present.

## Figures and Tables

**Figure 1 genes-14-01592-f001:**
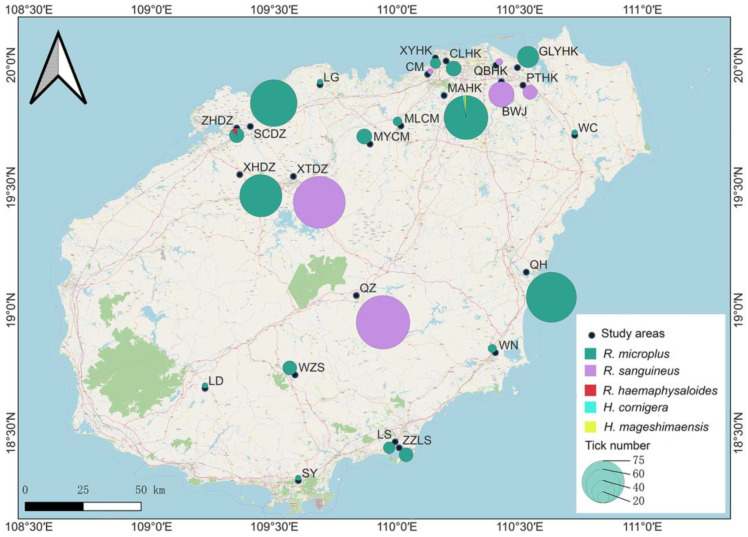
Collection sites of *Rhipicephalus* and *Haemaphysalis* ticks in Hainan Island, with tick numbers and species recorded at each study area. BWJ: Beiwuji, Meilan District; CLHK: Changliu Town, Xiuying District; CM: Chengmai County; GLYHK: Guilinyang, Meilan District; LD: Ledong Li Autonomous County; LS: Lingshui Li Autonomous County; MAHK: Meian Town, Xiuying District; MLCM: Meilang Village, Chengmai County; MYCM: Meiyang Village, County; PTHK: Potousan Village, Meilan District; QBHK: Qibi Village, Meilan District; QH: Qionghai City; QZ: Qiongzhong Li and Miao Autonomous County; SCDZ: Shancun Town, Danzhou City; SY: Sanya City; WN: Wanning City; WZS: Wuzhishan City; XHDZ: Xihuacha, Danzhou City; XTDZ: Xitian Village, Danzhou City; XYHK: Xiangtang Village, Xiuying District; ZHDZ: Zhonghe Town, Danzhou City; ZZLS: Zaozai Town, Lingshui Li Autonomous County.

**Figure 2 genes-14-01592-f002:**
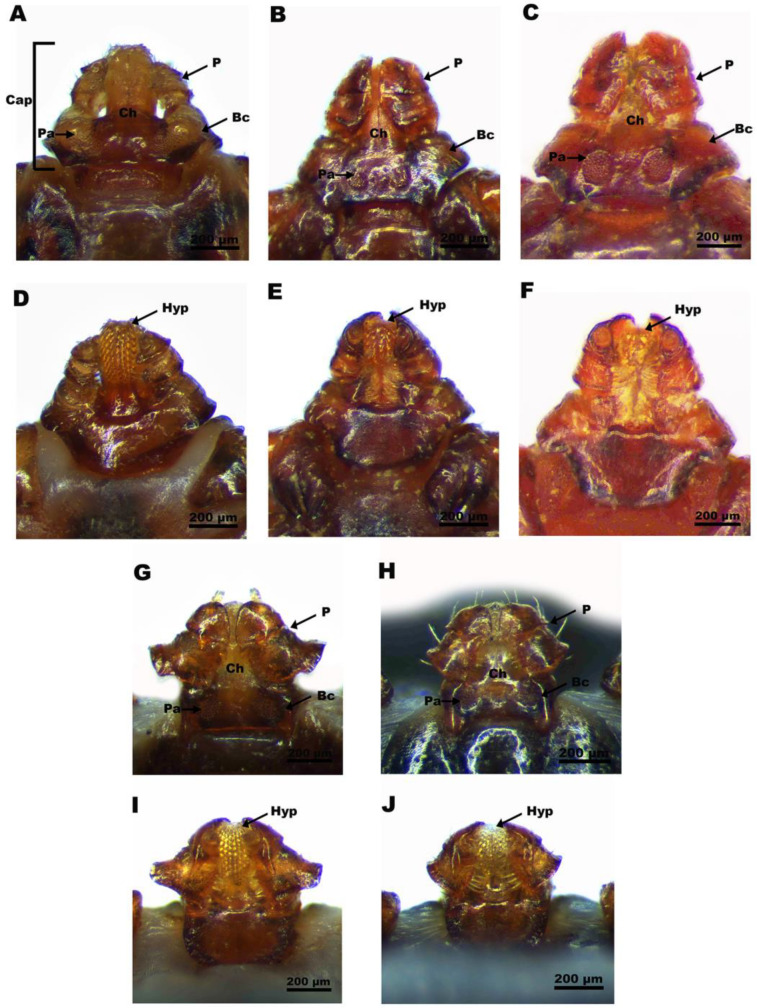
Morphological features of the capitulum of the adult female. *R. microplus* (**A**,**D**), *R. sanguineus (R. linnaei)* (**B**,**E**), *R. haemaphysaloides* (**C**,**F**), *H. cornigera* (**G**,**I**) and *H. mageshimaensis* (**H**,**J**) ticks collected in Hainan Island. (**A**–**C**,**G**,**H**): Dorsal view; (**D**–**F**,**I**,**J**): Ventral view. Bc, basis capitulum; Cap, capitulum; Ch, chelicerae; Hyp, hypostome; P, palps; Pa, porose area.

**Figure 3 genes-14-01592-f003:**
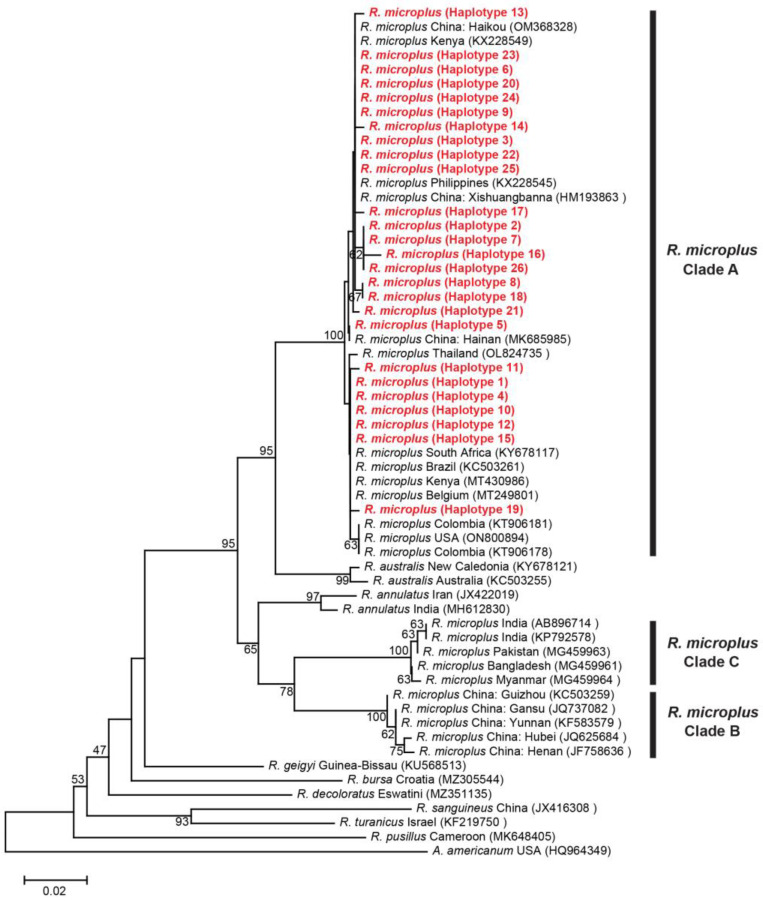
Neighbor-joining (NJ) phylogenetic tree of *R. microplus* based on the *cox1* gene sequences. Evolutionary analyses were conducted in MEGA XI. Bootstrap values (1000 replications) are shown on the branches. Sequences generated from this study are in red label.

**Figure 4 genes-14-01592-f004:**
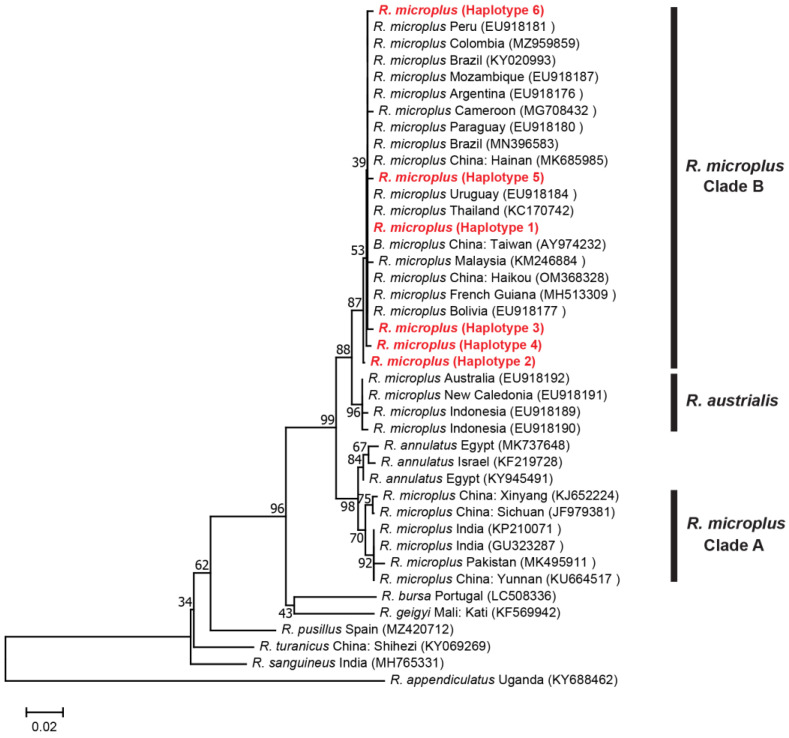
Neighbor-joining (NJ) phylogenetic tree of *R. microplus* based on the *16S* rRNA gene sequences. Evolutionary analyses were conducted in MEGA XI. Bootstrap values (1000 replications) are shown on the branches. Sequences generated from this study are in red label.

**Figure 5 genes-14-01592-f005:**
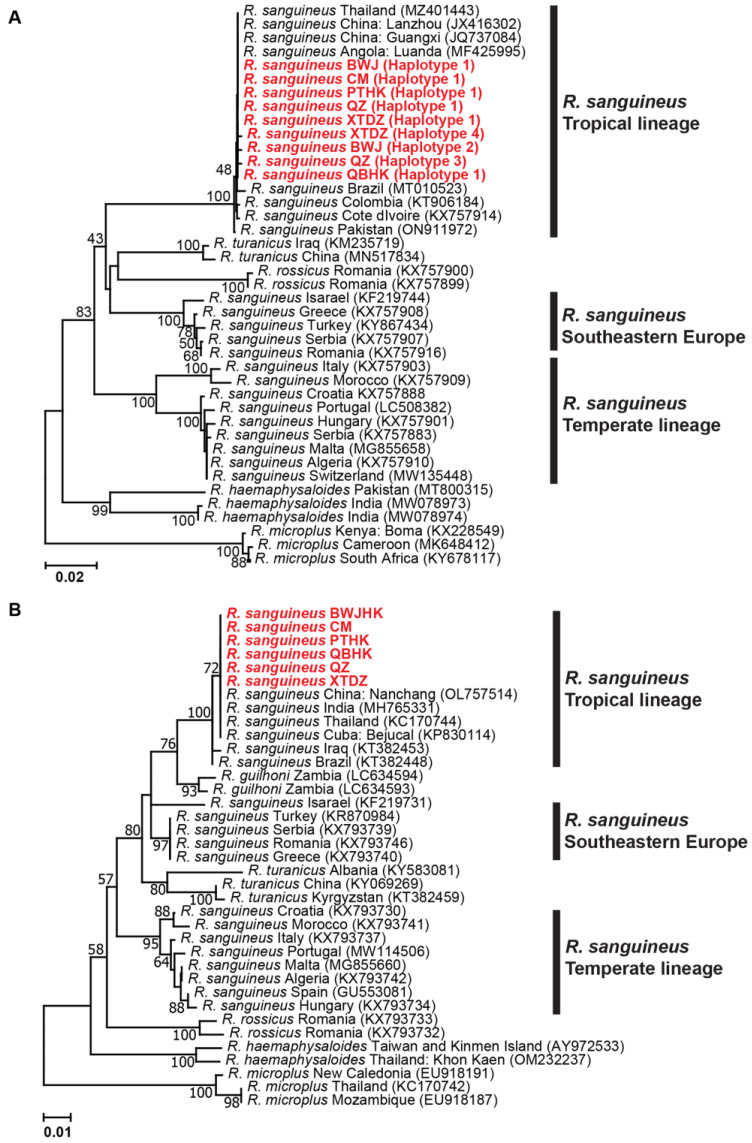
Neighbor-joining (NJ) phylogenetic tree of *R. sanguineus* based on the *cox1* gene (**A**) and *16S* rRNA gene (**B**) sequences. Evolutionary analyses were conducted in MEGA XI. Bootstrap values (1000 replications) are shown on the branches. Sequences generated from this study are in red label.

**Figure 6 genes-14-01592-f006:**
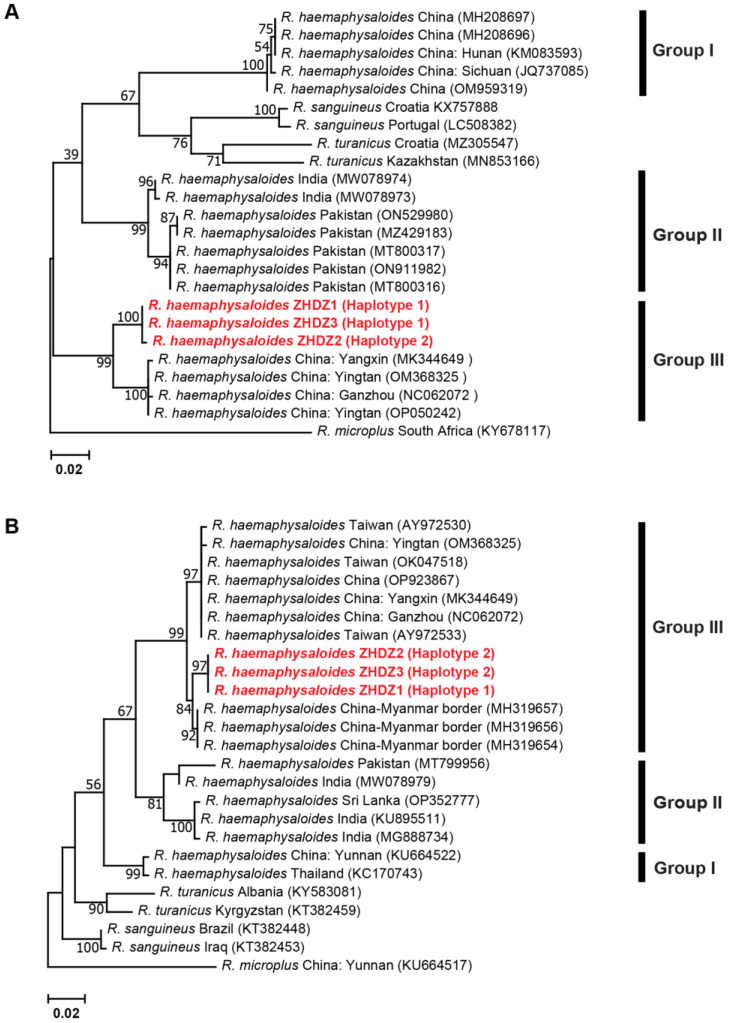
Maximum Likelihood (ML) phylogenetic tree of *R. haemaphysaloides* based on the *cox1* gene (**A**) and *16S* rRNA gene (**B**) sequences. Evolutionary analyses were conducted in MEGA XI. Bootstrap values (1000 replications) are shown on the branches. Sequences generated from this study are in red label.

**Figure 7 genes-14-01592-f007:**
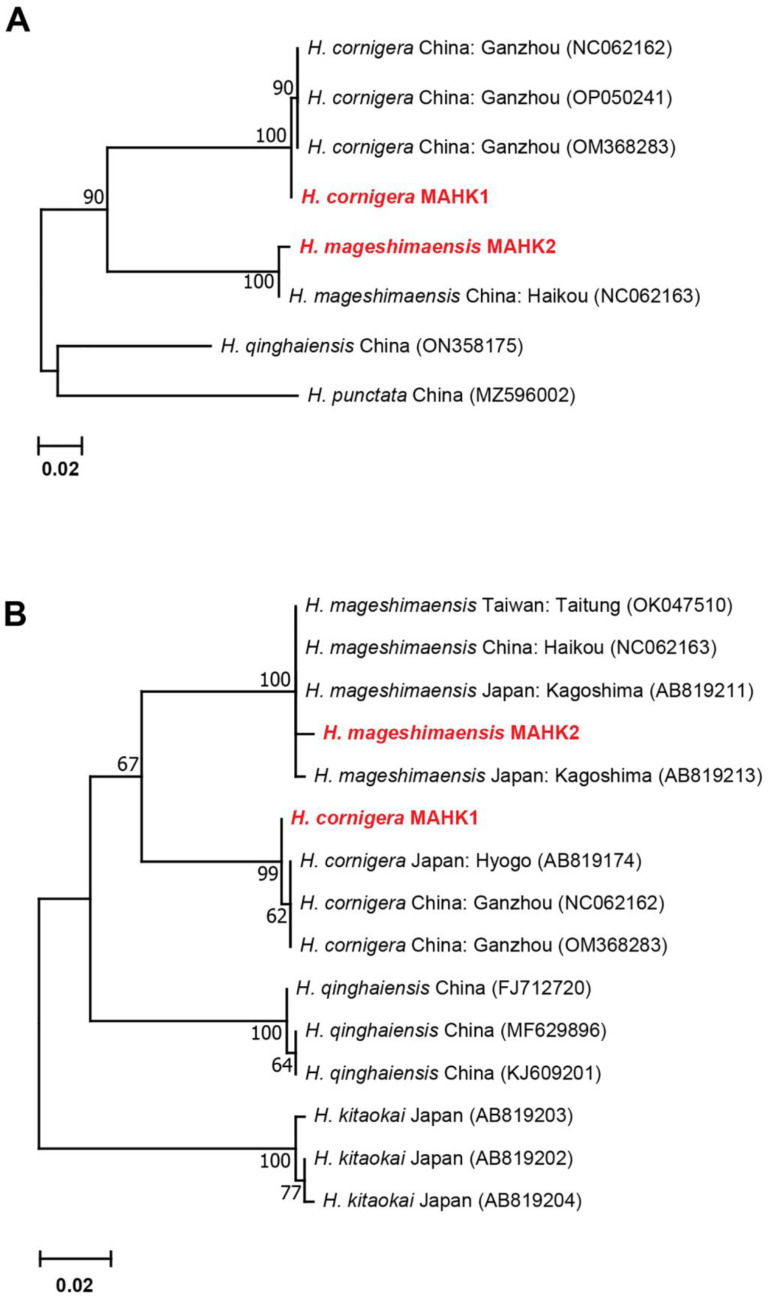
Maximum Likelihood (ML) phylogenetic tree of *H. cornigera* and *H. mageshimaensis* based on the *cox1* gene (**A**) and *16S* rRNA gene (**B**) sequences. Evolutionary analyses were conducted in MEGA XI. Bootstrap values (1000 replications) are shown on the branches. Sequences generated from this study are in red label.

## Data Availability

The genetic information obtained in this study is currently being submitted to the National Center for Biotechnology Information (NCBI).

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
