# Peer review of "Morphological and Molecular Identification of Hard Ticks in Hainan Island, China"

_genes, 2023, doi:10.3390/genes14081592_

Round 1
Reviewer 1 Report
The authors investigated prevalence and taxonomic diversity of hard ticks in Hainan Island, China based on very impressive large data set (more than 800 specimens were analysed morphologically and almost 200 were analysed genetically). The topic of the paper is very interesting; it represents novel observations of actual problematic. The text is well organized and well written and I encourage its acceptance for publication. I do not see much problems there.
However, the following sentence in Abstract (lines 26-27) somewhat confusing and not clear, especially for readers not familiar with the group in question. “R. sanguineus was recognized as a member of the tropical lineage by phylogenetic analysis on the cox1 and 16S rRNA genes, and therefore named R. linnaei.”
I would recommend providing short explanation (e.g. in the main text), that due to genetic distinctness from temperate R. sanguineus, tropical lineage are considered in the literature by some authors as distinct taxon R. linnaei.
Also, I would recommend to avoid constant usage of following form “R. sanguineus sensu lato (s.l.) tropical lineage (or R. linnaei)”. As for me, it is better to state once, that as soon as R. sanguineus from Hainan Island genetically refers to the tropical lineage (not to the temperate one) of R. sanguineus, known from the literature as R. linnnaei, we will use hereafter name R. linnaei for Hainan populations of R. sanguineus.
At least, do not use (or. R. linnaei), since it might be considered as opposition… or R. sanguineus or R. linnaei. R. sanguineus (R. linnaei) is fine.
The authors investigated prevalence and taxonomic diversity of hard ticks in Hainan Island, China based on very impressive large data set (more than 800 specimens were analysed morphologically and almost 200 were analysed genetically). The topic of the paper is very interesting; it represents novel observations of actual problematic. The text is well organized and well written and I encourage its acceptance for publication. I do not see much problems there.
However, the following sentence in Abstract (lines 26-27) somewhat confusing and not clear, especially for readers not familiar with the group in question. “R. sanguineus was recognized as a member of the tropical lineage by phylogenetic analysis on the cox1 and 16S rRNA genes, and therefore named R. linnaei.”
I would recommend providing short explanation (e.g. in the main text), that due to genetic distinctness from temperate R. sanguineus, tropical lineage are considered in the literature by some authors as distinct taxon R. linnaei.
Also, I would recommend to avoid constant usage of following form “R. sanguineus sensu lato (s.l.) tropical lineage (or R. linnaei)”. As for me, it is better to state once, that as soon as R. sanguineus from Hainan Island genetically refers to the tropical lineage (not to the temperate one) of R. sanguineus, known from the literature as R. linnnaei, we will use hereafter name R. linnaei for Hainan populations of R. sanguineus.
At least, do not use (or. R. linnaei), since it might be considered as opposition… or R. sanguineus or R. linnaei. R. sanguineus (R. linnaei) is fine.
English is fine, minor corrections required
Author Response
Responses to reviewer’ comments for the manuscript (Manuscript ID: genes-2518785) entitled “Morphological and molecular identification of hard ticks in Hainan Island, China”
Reviewer #1:
We would like to thank the reviewer for the thoughtful comments and suggestions, which are helpful in improving the manuscript. We have revised our manuscript accordingly. The changes in the revised manuscript are in accordance with the reviewer’s suggestions and are shown in the marked copy (highlighted in yellow). Point-by-point explanations for all of the revisions are listed below.
Comments and Suggestions for Authors
The authors investigated prevalence and taxonomic diversity of hard ticks in Hainan Island, China based on very impressive large data set (more than 800 specimens were analysed morphologically and almost 200 were analysed genetically). The topic of the paper is very interesting; it represents novel observations of actual problematic. The text is well organized and well written and I encourage its acceptance for publication. I do not see much problems there.
Q1: However, the following sentence in Abstract (lines 26-27) somewhat confusing and not clear, especially for readers not familiar with the group in question. “R. sanguineus was recognized as a member of the tropical lineage by phylogenetic analysis on the cox1 and 16S rRNA genes, and therefore named R. linnaei.”
A1: In accordance with the reviewer’s suggestion, and in order not to confuse the reader, the sentence has been edited as follows: “Rhipicephalus sanguineus was recognized as a member of the tropical lineage by phylogenetic analysis on the cox1 and 16S rRNA genes” (Page 1, line 28-29).
Q2: I would recommend providing short explanation (e.g. in the main text), that due to genetic distinctness from temperate R. sanguineus, tropical lineage are considered in the literature by some authors as distinct taxon R. linnaei.
A2: A short explanation has been provided according to the reviewer’s suggestion as follows: “Šlapeta et al. (2021) confirmed that Rhipicephalus linnaei, which is synonymous with R. sanguineus, was regarded as the oldest name and suggested that its adoption should refer to R. sanguineus sensu lato as the "tropical lineage”. In addition, Šlapeta et al. (2022) elucidated further regarding recognition of the tropical lineage (R. sanguineus s. l.) and officially designated it as Rhipicephalus linnaei.” (Page 11, line 329-333).
References:
Šlapeta, J., Chandra, S., Halliday, B. The “tropical lineage” of the brown dog tick Rhipicephalus
sanguineus sensu lato identified as Rhipicephalus linnaei (Audouin, 1826). Int. J. Parasitol 2021, 51, 431–436.
Šlapeta, J., Halliday, B., Chandra, S., Alanazi, A.D., Abdel-Shafy, S. Rhipicephalus linnaei
(Audouin, 1826) recognised as the "tropical lineage" of the brown dog tick Rhipicephalus sanguineus sensu lato: Neotype designation, redescription, and establishment of morphological and molecular reference. Ticks Tick Borne Dis 2022, 13, 102024.
Liang, D., Chen, H., An, L., Li, Y., Zhao, P., Upadhyay, A., Hansson, B.S., Zhao, J., Han, Q.
Molecular identification and functional analysis of Niemann-Pick type C2 proteins,carriers for semiochemicals and other hydrophobic compounds in the brown dog tick, Rhipicephalus linnaei. Pestic Biochem Physiol 2023, 193, 105451.
Q3: Also, I would recommend to avoid constant usage of following form “R. sanguineus sensu lato (s.l.) tropical lineage (or R. linnaei)”. As for me, it is better to state once, that as soon as R. sanguineus from Hainan Island genetically refers to the tropical lineage (not to the temperate one) of R. sanguineus, known from the literature as R. linnaei, we will use hereafter name R. linnaei for Hainan populations of R. sanguineus.
A3” This has been edited according to the reviewer’s suggestion. We have avoided constant usage of the following form, “R. sanguineus sensu lato (s.l.) tropical lineage (or R. linnaei)” throughout the manuscript and renamed it R. sanguineus (R. linnaei).
Q4: At least, do not use (or. R. linnaei), since it might be considered as opposition… or R. sanguineus or R. linnaei. R. sanguineus (R. linnaei) is fine.
A4: This has been carried out according to the reviewer’s suggestion. We have renamed R. sanguineus (or R. linnaei) to R. sanguineus sensu lato (s.l.) tropical lineage (R. linnaei) for the first time and thereafter used R. linnaei for Hainan populations of R. sanguineus.

Reviewer 2 Report
Dear Editor,
The manuscript is well written and I have little to observe. I personally do not agree much with the use of the COI gene for phylogenetic analysis, but only for dentification or species discrimination. Among other things, no real phylogenetic analysis has been done, even though the term is repeatedly used perhaps excessively. For phylogenetic analyses, 16S rRNA is more suitable and more emphasis should be placed on these analyses. I also suggest the use of ASAP (https://bioinfo.mnhn.fr/abi/public/asap) as a method of species delimitation using the COI gene.

Author Response
Responses to reviewer’ comments for the manuscript (Manuscript ID: genes-2518785) entitled “Morphological and molecular identification of hard ticks in Hainan Island, China”
Reviewer #2:
Comments and Suggestions for Authors
Dear Editor,
Q1: The manuscript is well written and I have little to observe. I personally do not agree much with the use of the COI gene for phylogenetic analysis, but only for identification or species discrimination. Among other things, no real phylogenetic analysis has been done, even though the term is repeatedly used perhaps excessively. For phylogenetic analyses, 16S rRNA is more suitable and more emphasis should be placed on these analyses. I also suggest the use of ASAP (https://bioinfo.mnhn.fr/abi/public/asap) as a method of species delimitation using the COI gene.
A1: Thank you for your valuable feedback and constructive suggestions on our manuscript. We appreciate your positive remarks about the overall quality of the writing. Regarding your concern about use of the COI gene for phylogenetic analysis, we understand your viewpoint and acknowledge the distinction between the COI's utility for species identification rather than true phylogenetic analysis. We apologize for any potential confusion arising from the repeated use of the term "phylogenetic analysis" in the manuscript. In response to your suggestion, we agree that using 16S rRNA for phylogenetic analysis would be more appropriate, and we have taken this into serious consideration during the revision process. By incorporating 16S rRNA data, we have provided a more robust and accurate phylogenetic analysis to enhance the reliability of our findings.
Regarding the use of ASAP for species delimitation using the COI gene, we acknowledge its potential benefits. We are committed to delivering a high-quality and comprehensive research study, and we assure you that we will consider implementing ASAP in our future research to further enhance the accuracy of species delimitation. Once again, we sincerely appreciate your input, which has been instrumental in improving the quality of our research. We used the ASAP (Assemble Species by Automatic Partitioning) program for phylogenetic analysis of 5 tick species, based on COI and 16S rRNA genes. The findings demonstrated that both the ASAP and MEGA programs produced comparable results for the classification test.
In addition, we would like to thank the reviewer for the thoughtful comments and suggestions, which are helpful in improving the manuscript. Based on these, we have made careful revisions to the original manuscript. The changes in the revised manuscript, according to the reviewer’s suggestions, are shown in the marked copy (highlighted in yellow). Detailed comments and explanations for the revisions are listed below.
Abstract.
We have moved sentence 1 to before line 22 according to the reviewer’s suggestions (Page 1, line 21-25).
Introduction.
This has been edited according to the reviewer’s suggestion as follows:
“Ticks are blood-sucking ectoparasitic arthropods that play an essential role in transmitting various pathogens to humans and animals worldwide, they are the second most significant transmitters of infectious diseases after mosquitoes” (Page 1, line 40-42).
“Morphological identification refers to the process of visually identifying tick species based on their morphological characteristics, such as body shape, size, color and pattern” (Page 2, line 49-51).
Results.
This has been edited according to the reviewer’s suggestion as follows:
We have rewritten R. microplus to the full name of “Rhipicephalus microplus” (Page 4, line 168).
We changed the word “Phylogenetic analysis” to “Genetic analysis” (Page 6, line 211).

Reviewer 3 Report
Review of Intirach et al Morphological….etc. in Hainan Island.
General comments:
This is a useful overview of the different tick species on Hainan island, how to identify them, and their genetic diversity in different locations throughout the island. There are minor issues that should be addressed as described below.
Abstract.
Renaming of R. sanguineus on Hainan Island to a full species based on the data provided is not justified. However, it could be assigned to a subspecies of R. sanguineus.
Introduction.
Line 53. Change complex to “complexes” (plural of complex).
Line 72. Add “features” after morphological to differentiate morphological from nucleotide analysis.
Materials and Methods
Line 129-130. No template control was observed? What does this mean? No template control was used, or No template control did not show any evidence of DNA? Please clarify.
Results
Lines 314-315. This table is difficult to interpret. It should be eliminated from the main text. It could be relegated to supplementary files if the authors wish to keep, but it is virtually meaningless in the main article.
Line 321. Figure 8 is meaningless for all but the most highly specialized reader, if at all. The authors have already established that there are numerous haplotypes among the population of R. sanguineus throughout Hainan island. Therefore, this figure should be deleted. If the authors still want to keep it, it could be relegated to the supplementary files.
Discussion
Lines 326 – 335. This first paragraph is not discussion. Instead, it is just a recapitulation of the introductory and results sections. Begin with paragraph 2, line 336.
Line 397. Frequent gene communication. Not clear what this means? Are you referring to genetic interchanges? Please clarify.
Conclusions.
The summary statement is not truly a conclusion, just an elementary recapitulation of the genetic diversity while ignoring the morphological basis of species identification, the number of species, their distribution throughout the island, etc. The authors need to broaden their conclusions to encompass the island’s tick population.
Please the file attached above Initrach et al...etc.
Author Response
Responses to reviewer’ comments for the manuscript (Manuscript ID: genes-2518785) entitled “Morphological and molecular identification of hard ticks in Hainan Island, China”
Reviewer #3:
Comments and Suggestions for Authors
Review of Intirach et al Morphological….etc. in Hainan Island.
General comments:
This is a useful overview of the different tick species on Hainan Island, how to identify them, and their genetic diversity in different locations throughout the island. There are minor issues that should be addressed as described below.
We would like to thank the reviewer for the thoughtful comments and suggestions, which are helpful in improving the manuscript. We have revised our manuscript accordingly. The changes in the revised manuscript are in accordance with the reviewer’s suggestions and are shown in the marked copy (highlighted in yellow). Point-by-point explanations for all of the revisions are listed below.
Abstract.
Q1: Renaming of R. sanguineus on Hainan Island to a full species based on the data provided is not justified. However, it could be assigned to a subspecies of R. sanguineus.
A1: Thank you for your advice. Šlapeta et al. (2021) confirmed that Rhipicephalus linnaei was synonymous with R. sanguineus and considered the oldest name, and they proposed that its adoption should refer to R. sanguineus sensu lato as the “tropical lineage”. In addition, the identity of R. sanguineus s. l., tropical lineage is clarified further and renamed Rhipicephalus linnaei by Šlapeta et al. (2022) (Qian et al., 2023). The tick we studied here was identified as the tropical lineage (R. sanguineus s. l.), now renamed R. linnaei, by using a partial 16S rRNA gene sequence and morphological data.
However, this has been edited according to the reviewer’s suggestion (Page 1, line 34).
References:
Šlapeta, J., Chandra, S., Halliday, B. The “tropical lineage” of the brown dog tick Rhipicephalus
sanguineus sensu lato identified as Rhipicephalus linnaei (Audouin, 1826). Int. J. Parasitol 2021, 51, 431–436.
Šlapeta, J., Halliday, B., Chandra, S., Alanazi, A.D., Abdel-Shafy, S. Rhipicephalus linnaei
(Audouin, 1826) recognised as the "tropical lineage" of the brown dog tick Rhipicephalus sanguineus sensu lato: Neotype designation, redescription, and establishment of morphological and molecular reference. Ticks Tick Borne Dis 2022, 13, 102024.
Liang, D., Chen, H., An, L., Li, Y., Zhao, P., Upadhyay, A., Hansson, B.S., Zhao, J., Han, Q.
Molecular identification and functional analysis of Niemann-Pick type C2 proteins,carriers for semiochemicals and other hydrophobic compounds in the brown dog tick, Rhipicephalus linnaei. Pestic Biochem Physiol 2023, 193, 105451.
Introduction.
Q2: Line 53. Change complex to “complexes” (plural of complex). Line 72. Add “features” after morphological to differentiate morphological from nucleotide analysis.
A2: This has been edited according to the reviewer’s suggestion as follows: “complex to “complexes” (Page 2, line 55), and “features” has been added after morphological (Page 2, line 74).
Materials and Methods
Q3: Line 129-130. No template control was observed? What does this mean? No template control was used, or No template control did not show any evidence of DNA? Please clarify.
A3: No template control means the omission of any DNA or RNA template from a reaction, and serves as a general control for extraneous nucleic acid contamination. The sentence has been rewritten according to the reviewer’s suggestion as follows: “A negative no-template water control was included in all of the PCR runs” (Page 3, line 131-132).
Results
Q4: Lines 314-315. This table is difficult to interpret. It should be eliminated from the main text. It could be relegated to supplementary files if the authors wish to keep, but it is virtually meaningless in the main article.
A4: Table 1 has been deleted according to the reviewer’s suggestion, and relegated to Supplementary Table S13.
Q5: Line 321. Figure 8 is meaningless for all but the most highly specialized reader, if at all. The authors have already established that there are numerous haplotypes among the population of R. sanguineus throughout Hainan Island. Therefore, this figure should be deleted. If the authors still want to keep it, it could be relegated to the supplementary files.
A5: Figure 8 has been deleted according to the reviewer’s suggestion, and relegated to Supplementary Figure S4.
Discussion
Q6: Lines 326 – 335. This first paragraph is not discussion. Instead, it is just a recapitulation of the introductory and results sections. Begin with paragraph 2, line 336.
A6: We agree with the reviewer, and the Discussion begins with paragraph 2 (Page 11, line 316).
Q7: Line 397. Frequent gene communication. Not clear what this means? Are you referring to genetic interchanges? Please clarify.
A7: The sentence has been rewritten according to the reviewer’s suggestion as follows: “It is indicated that ticks may have the ability to adapt to different environments and there was frequent gene communication between individuals or populations” (Page 12, line 380-381).
Q8: Conclusions.
The summary statement is not truly a conclusion, just an elementary recapitulation of the genetic diversity while ignoring the morphological basis of species identification, the number of species, their distribution throughout the island, etc. The authors need to broaden their conclusions to encompass the island’s tick population.
A8: The statement has been rewritten according to the reviewer’s suggestion as follows: “This study employed morphological and molecular approaches to characterize and identify genetic relationships of the field-collected ticks from eight hundred and fifty-eight hard ticks collected from cattle, dogs and goats in 24 locations around Hainan Island. Based on morphological features, five tick species were identified, namely Rhipicephalus microplus, R. sanguineus (R. linnaei), R. haemaphysaloides, Haemaphysalis cornigera and H. mageshimaensis, and confirmed subsequently by molecular analysis of cox1 and 16S rRNA genes. It was concluded that cox1 and 16S rRNA genes were useful markers for verifying species identification of hard ticks, according to analyses of the sequence data obtained. The analyses established a dependable DNA reference database that could be utilized in forensic entomology not only in China, but also in other countries where these species are present.” (Page 12, lines 385-395).
